# Recombinant γY278H Fibrinogen Showed Normal Secretion from CHO Cells, but a Corresponding Heterozygous Patient Showed Hypofibrinogenemia

**DOI:** 10.3390/ijms22105218

**Published:** 2021-05-14

**Authors:** Tomu Kamijo, Takahiro Kaido, Masahiro Yoda, Shinpei Arai, Kazuyoshi Yamauchi, Nobuo Okumura

**Affiliations:** 1Department of Medical Sciences, Graduate School of Medicine, Science and Technology, Shinshu University, Matsumoto 390-8621, Japan; kamtom@shinshu-u.ac.jp (T.K.); 21hm207d@shinshu-u.ac.jp (T.K.); 2Department of Laboratory Medicine, Shinshu University Hospital, Matsumoto 390-8621, Japan; 3Department of Clinical Laboratory Investigation, Graduate School of Medicine, Shinshu University, Matsumoto 390-8621, Japan; masaru25250308@gmail.com (M.Y.); arais@shinshu-u.ac.jp (S.A.); yamauchi@shinshu-u.ac.jp (K.Y.); 4Laboratory of Clinical Chemistry and Immunology, Department of Biomedical Laboratory Sciences, School of Health Sciences, Shinshu University, Matsumoto 390-8621, Japan

**Keywords:** congenital fibrinogen disorders, hypofibrinogenemia, Chinese hamster ovary cells, recombinant fibrinogen, plasmin degradation

## Abstract

We identified a novel heterozygous hypofibrinogenemia, γY278H (Hiroshima). To demonstrate the cause of reduced plasma fibrinogen levels (functional level: 1.12 g/L and antigenic level: 1.16 g/L), we established γY278H fibrinogen-producing Chinese hamster ovary (CHO) cells. An enzyme-linked immunosorbent assay demonstrated that synthesis of γY278H fibrinogen inside CHO cells and secretion into the culture media were not reduced. Then, we established an additional five variant fibrinogen-producing CHO cell lines (γL276P, γT277P, γT277R, γA279D, and γY280C) and conducted further investigations. We have already established 33 γ-module variant fibrinogen-producing CHO cell lines, including 6 cell lines in this study, but only the γY278H and γT277R cell lines showed disagreement, namely, recombinant fibrinogen production was not reduced but the patients’ plasma fibrinogen level was reduced. Finally, we performed fibrinogen degradation assays and demonstrated that the γY278H and γT277R fibrinogens were easily cleaved by plasmin whereas their polymerization in the presence of Ca^2+^ and “D:D” interaction was normal. In conclusion, our investigation suggested that patient γY278H showed hypofibrinogenemia because γY278H fibrinogen was secreted normally from the patient’s hepatocytes but then underwent accelerated degradation by plasmin in the circulation.

## 1. Introduction

Fibrinogen is a 340-kDa plasma glycoprotein composed of three different polypeptide chains (Aα: 610, Bβ: 461, and γ: 411 residues) [1], which are encoded by three genes designated *FGA*, *FGB*, and *FGG*, respectively. These chains are synthesized, assembled into a three-chain monomer (Aα-Bβ-γ), and held together into a six-chain dimer (Aα-Bβ-γ)_2_ by 29 disulfide bonds in hepatocytes [2]. The dimer is then secreted into circulation at a concentration of 1.8–3.5 g/L in plasma. The molecule has a linear structure with a central E-region and two partial D-regions connected with a coiled-coil. The E-region contains the *N*-termini of all chains, and the D-regions contain the C-termini of the Bβ (β-module) and γ chains (γ-module) in addition to a short segment of the Aα chains [3,4].

All numbers of amino acid residues are presented with reference to the mature protein in plasma. The conversion of fibrinogen to fibrin is the final step in the blood coagulation cascade and is essential for hemostasis and thrombosis. At the initiation of fibrin formation, fibrinogen is cleaved by thrombin and converts to fibrin monomers, which polymerize spontaneously. First, the “A-a” knob-hole interactions mediate the formation of double-stranded protofibrils with a half-staggered overlap between molecules in different strands [5]. In this step, each protofibril requires the so-called “D-D” interaction. The “D-D” interface composed of γR275 to γM310, especially γR275, γY280, and γS300, in the γ-module and abuts the γ chain of two adjacent molecules [6,7]. These protofibrils grow in length to a 20–25-mer oligomer and the “B-b” knob-hole interaction promotes lateral aggregation [8], resulting in the formation of thicker fibers and an insoluble fibrin clot consisting of a multi-stranded and branched fiber network [9].

Other than the “D-D” interface, the fibrinogen γ-module contains many functional sites and structures: hole ‘a’; γ-γ cross-linking; and high affinity Ca^2+^-, αM/β2-, GPIIb/IIIa-, and tissue plasminogen activator-binding sites [10,11]. Therefore, fibrinogens with alterations in the γ-module are important tools for examining the normal function of the fibrinogen to fibrin conversion.

More than 1200 inherited fibrinogen variants have been reported so far [12,13]. These genetic mutations in *FGA*, *FGB*, and *FGG* have been associated with phenotypes of either hypofibrinogenemia or dysfibrinogenemia. Hypofibrinogenemia, including afibrinogenemia, has the quantitative characteristic level of fibrinogen and has been defined as reduced functional and antigenic fibrinogen levels in plasma. Dysfibrinogenemia has the qualitative characteristic level of fibrinogen and is defined as reduced functional but with normal antigenic levels in plasma. Hypodysfibrinogenemia has the characteristics of hypofibrinogenemia and dysfibrinogenemia [14].

Here, we describe a novel heterozygous fibrinogen variant, γY278H, which manifested hypofibrinogenemia. To demonstrate why the plasma fibrinogen level was reduced in this patient, we produced γY278H fibrinogen-producing Chinese hamster ovary (CHO) cells and analyzed the synthesis and secretion of variant fibrinogen. Fibrinogen secretion by CHO cells was almost normal; therefore, we produced additional variant fibrinogen-producing CHO cells, γL276P, γT277P, γT277R, γA279D and γY280C, and performed further investigations.

## 2. Results

### 2.1. Patient’s Coagulation Tests, DNA Sequence Analysis and Immunoblotting Analysis

The patient was a 31-year-old Japanese woman who had a miscarriage and bleeding after the delivery of her first child, but other bleeding or thrombotic complications were not observed. Her prothrombin time (PT), activated partial thromboplastin time (APTT), D-dimer, and fibrin/fibrinogen degradation products (FDPs) were 11.6 s (normal range: 10.8–13.2 s), 37.1 s (normal range: 22.5–37.5 s), <0.1 mg/L (normal range: <1.0 mg/L), and <2.5 mg/L (normal range: <5.0 mg/L), respectively. Her functional fibrinogen level was 1.12 g/L and antigenic fibrinogen level was 1.16 g/L (normal range: 1.80–3.50 g/L).

DNA sequence analysis revealed a heterozygous mutation of T > C in *FGG* exon 8, resulting in the substitution of Tyr (TAT) for His (CAT) at residue γ278 (mature protein; residue γ304 in the native protein). Because this mutation was the first case in the world, we designated this patient as Hiroshima (γY278H) according to the patient’s place of residence.

Immunoblotting analysis of the patient’s plasma revealed that the mobility of the fibrinogen, γ-γ dimer and Aα, Bβ, and γ chains in the patient plasma were similar to those of normal plasma and no extra bands were detected (Figure 1).

### 2.2. Secretion and Synthesis of Variant Fibrinogens in CHO Cells

Wild-type and variant fibrinogens were expressed in CHO cells, and fibrinogen concentrations in the culture media (amount of secretion) and cell lysates (amount of synthesis) were measured using an enzyme-linked immunosorbent assay (ELISA) (Figure 2). The fibrinogen concentration of wild-type (*n* = 11) was 0.76 (0.47–0.90) µg/mL (median (interquartile range)) in culture media and 0.64 (0.34–0.78) μg/mL in cell lysates, resulting in a ratio in culture media to cell lysates (M/C ratio) of 1.21 (0.98–1.39). The fibrinogen concentrations of γY278H fibrinogen-producing cell lines (*n* = 11) were 3.02 (2.23–4.60) μg/mL in culture media and 4.68 (3.34–6.15) μg/mL in cell lysates, and the M/C ratio was 0.75 (0.54–0.82), which was lower than that of wild-type (*p* < 0.01), but both synthesis and secretion were not completely reduced. The M/C ratios of γL276P [*n* = 11, 0.06 (0.04–0.08)] and γT277P [*n* = 12, 0.03 (0.02–0.04)] fibrinogen-producing cell lines were significantly lower (*p* < 0.001) because they had higher amounts of synthesis and lower amounts of secretion than those of wild-type. The M/C ratio of γY280C [*n* = 10, 0.79 (0.64–0.97)] fibrinogen-producing cell lines was slightly lower (*p* < 0.05), but those of γT277R [*n* = 11, 2.06 (1.54–2.39)] and γA279D [*n* = 10, 1.51 (1.28–1.85)] fibrinogen-producing cell lines were slightly higher than that of wild-type (*p* < 0.05 and non-significant, respectively).

To confirm the synthesis of variant γ chains and the assembly of fibrinogen, cell lysates of each variant fibrinogen-producing cell line were analyzed by immunoblotting (Figure 3). The analysis demonstrated that variant γ chains and fibrinogens were synthesized in all variant cell lines, and their mobilities were similar to the wild-type cell line. As shown in Figure 3C, the γ-γ dimer levels of γY278H and γT277R were observed clearly, but those in γL276P and γY280C were reduced and in γT277P and γA279D were hardly observed.

### 2.3. Fibrinogen Degradation Assay Using Plasmin

To assess the level of fibrinogen degradation by plasmin of γY278H and γT277R, whose fibrinogen-producing cell lines secrete enough fibrinogen to purify, we performed a fibrinogen degradation assay in the presence of 5 mM ethylenediaminetetraacetic acid (EDTA). Wild-type was used as a normal control and γR275C, in which a markedly altered D:D interaction was reported [15], was used as a dysfunctional variant control. It has been reported that fibrinogen is cleaved into fragment D1 by plasmin, and fragment D1 is further cleaved into fragments D2 and D3 [16]. As shown in Figure 4A, recombinant wild-type fibrinogen was cleaved into fragments D1 and D2 by a 0.5-h incubation with plasmin, and the amount of fragment D1 decreased, whereas those of fragments D2 and D3 increased with a longer incubation. In γY278H fibrinogen, fragment D1 was more readily cleaved into fragment D2 than that of wild-type after a 0.5-h incubation, and fragments D1, D2, and D3 were further cleaved into approximately 75 kDa fragments, which we designated as fragment D4, and lower molecular weight fragments with longer incubation (Figure 4D). Although the amount of fragment D1 formed from γT277R fibrinogen after a 0.5-h incubation was less than that of γY278H, fragments D1, D2, and D3 were also cleaved into fragment D4 and further cleaved into lower molecular weight fragments with longer incubation (Figure 4C). On the other hand, fragment D1 of γR275C after 0.5- and 1-h incubations was more cleaved than wild-type but fragment D4 and lower molecular weight fragments were hardly observed (Figure 4B).

### 2.4. Thrombin-Catalyzed Fibrin Polymerization

To assess the fibrinogen function of γY278H and γT277R, we performed a thrombin-catalyzed fibrin polymerization, and representative curves of triplicate experiments are shown in Figure 5. In the presence of Ca^2+^, recombinant γY278H and γT277R fibrinogens had a similar lag time (3.5 ± 0.3 min and 2.6 ± 0.3 min, respectively) and maximum slope (81.5 ± 31.1 × 10^−5^/s and 135.8 ± 14.7 × 10^−5^/s, respectively; *p* < 0.01) compared with wild-type (2.5 ± 0.2 min and 98.6 ± 17.5 × 10^−5^/s). For γR275C fibrinogen, no increase in turbidity was found, as reported previously [15]. In the absence of Ca^2+^, γY278H fibrinogen had a four-fold longer lag time (13.6 ± 3.3 min, *p* < 0.001) and a four-fold gentler slope (22.8 ± 5.9 × 10^−5^/s, *p* < 0.001) compared with those in the presence of Ca^2+^, but γT277R showed a similar lag time (1.8 ± 0.1 min) and maximum slope (179.7 ± 14.5 × 10^−5^/s, *p* < 0.05) to those in the presence of Ca^2+^.

### 2.5. Protection Assay for the Plasmin Digestion of Fibrinogen

To assess the function of the low affinity Ca^2+^ binding site, we performed a protection assay for plasmin digestion in the presence of EDTA or Ca^2+^. As shown in Figure 6A, in the presence of 5 mM EDTA, fragment D1 derived from recombinant wild-type fibrinogen was cleaved into smaller fragments D2 and D3, indicating no protection from plasmin digestion. On the other hand, in the presence of 1 or 5 mM Ca^2+^, almost all of fragment D1 remained uncleaved, indicating strong protection against plasmin digestion.

For γY278H fibrinogen, fragment D1 was cleaved into fragments D2 and D3 and lower molecular weight fragments not only in the presence of EDTA, but also in the presence of Ca^2+^ (Figure 6D). Moreover, all remaining fragments were less than those of wild-type. For γT277R fibrinogen, fragments D2 and D3 were cleaved in a similar way to those of γY278H in the presence of EDTA, but fragment D1 was similar to that of wild-type fibrinogen in the presence of Ca^2+^ (Figure 6C). On the other hand, fragments D1, D2, and D3 of γR275C fibrinogen were similar to those of wild-type fibrinogen both in the presence of EDTA or Ca^2+^ (Figure 6B).

### 2.6. Factor (F) XIIIa-Catalyzed Cross-Linking of Fibrinogen

To assess the function of the “D-D” interactions of fibrinogen, we performed FXIIIa-catalyzed cross-linking of γ chains. Figure 7A shows that the cross-linked γ-γ dimer and α-polymer bands from recombinant wild-type fibrinogen were evident at 0.5 h. With a longer incubation, the intensity increased whereas that of γ and Aα chain bands decreased after each incubation period. With γY278H and γT277R fibrinogen, the γ-γ dimer and α-polymer bands appeared after 0.5 h and increased gradually after a longer incubation, similar to wild-type fibrinogen (Figure 7C,D). On the other hand, because γR275C fibrinogen has a markedly impaired “D:D” interaction, the γ-γ dimer and α-polymer bands at 0.5 h were slightly observed and did not increase with longer incubation compared with those of wild-type fibrinogen (Figure 7B).

## 3. Discussion

We identified the novel heterozygous variant, γY278H, which was designated as fibrinogen Hiroshima. γY278 is located at the “D:D” interface in the γ-module [7] and the site nearby is associated with fibrinogen secretion and function, because γR275C [17], γR275H [18], γA279D [19], and γY280C [20] have been reported as dysfibrinogenemia, γT277P [21] and γT277R [22] have been reported as hypofibrinogenemia.

γY278H patient’s antigenic fibrinogen level was 1.16 g/L and the functional/antigenic fibrinogen ratio was 0.97, resulting in the patient being categorized as having hypofibrinogenemia. In general, the M/C ratio of fibrinogen-producing cells transfected with hypofibrinogenemic variant plasmid was markedly reduced, but that of our established γY278H fibrinogen-producing cell line was approximately 60% of the wild-type fibrinogen-producing cell line. Therefore, the patient was expected to exhibit dysfibrinogenemia or hypodysfibrinogenemia, but these phenotypes were different in the actual patient’s phenotype, which was evaluated using coagulation tests.

To compare fibrinogen production with γY278H fibrinogen-producing cells, we established five additional variant fibrinogen-producing CHO cell lines, γL276P, γT277P, γT277R, γA279D, and γY280C. Apart from γL276P, which has not been reported as a congenital fibrinogen disorder, only the γT277R fibrinogen-producing cell line showed disagreement between the recombinant fibrinogen production phenotype and the actual patient’s phenotype, namely, the patient’s phenotype was hypofibrinogenemia (antigenic fibrinogen level 0.79 g/L and functional/antigenic fibrinogen ratio 0.90) but the predicted phenotype from recombinant fibrinogen production was dysfibrinogenemia or hypodysfibrinogenemia.

We have already established 33 γ-module variant fibrinogen-producing CHO cell lines, including 6 cell lines in this report, and have summarized the recombinant fibrinogen production ability and compared these with patients’ plasma antigenic fibrinogen levels (Table 1). We found that recombinant fibrinogen production from CHO cells was reduced when the M/C ratio was less than 0.5, and plasma fibrinogen levels were reduced when their antigenic fibrinogen levels were less than 1.5 g/L.

Three of 33 variant fibrinogen-producing CHO cell lines, γT277R, γY278H, and γT305A, showed disagreement, namely, the recombinant fibrinogen production was not reduced but the patient’s plasma fibrinogen level was reduced. However, γT305A patient’s plasma fibrinogen level might be lower than the normal range for adults, regardless of the patient’s genetic mutation, because the patient was a 6-month-old baby [46,47]. Therefore, marked inconsistency between recombinant fibrinogen production and plasma fibrinogen level was observed only in γT277R and γY278H. These results suggested that the recombinant fibrinogen producing ability of the γ chain variant using the CHO cell lines comprehensively reflected the plasma fibrinogen level, namely, the fibrinogen-producing ability of human hepatocytes. Moreover, Asselta (γT326H) [21], Duga (γG284R) [48], and Plate (γN345S) [49] used COS-1 cells, and Vu (γW227C) [50] used COS-7 cells for fibrinogen production and all of them reported that recombinant variant fibrinogen production reflected plasma fibrinogen levels. Therefore, our observations suggested that γY278H and γT277R patients’ hepatocytes synthesize and secrete γT277R or γY278H variant fibrinogen in their circulation.

One of the main enzymes of fibrinogen degradation in vivo is plasmin. Plasmin cleaves fibrinogen into one E- and two D-regions, and the D-region is then progressively cleaved into fragments D1, D2, and D3 [16]. Fragment D1 consists of the Aα105Asp-206Lys, Bβ134Asp-449Lys, and γ63Ala-406Lys, fragment D2 consists of the same Aα and Bβ chain and γ63Ala-356Lys, and fragment D3 consists of the same Aα and Bβ chain and γ63Ala-302Lys [31]. Fibrinogen degradation assays with plasmin indicated that recombinant γY278H and γT277R fibrinogens not only underwent faster cleavage into fragments D2 and D3 than wild-type and γR275C fibrinogens but also further cleavage into fragment D4 and lower molecular weight fragments, which were not observed in wild-type and γR275C. We predicted that fragment D4 consists of the same Aα and Bβ chain as fragment D3 but a shorter γ chain. Therefore, our observations suggested that γY278H and γT277Y patients’ plasma fibrinogens underwent accelerated degradation and induced the reduction of their plasma fibrinogen level, yet patients’ hepatocytes produced normal amount of variant fibrinogen. Because the molecular weights of plasmin degradation products were markedly lower and these were easier to cleave than fibrinogen, no extra bands were observed in immunoblotting analysis of patient’s plasma fibrinogen.

Finally, we investigated the functions of recombinant γT277R and γY278H fibrinogen. γT277R and γY278H fibrinogens showed almost normal polymerization in the presence of Ca^2+^ and normal “D:D” interaction. In the absence of Ca^2+^, γY278H fibrinogens showed reduced fibrin polymerization due to the impaired low affinity Ca^2+^ binding sites. However, in the patient’s circulation, γY278H plasma fibrinogen functioned normally because there was sufficient Ca^2+^ present in plasma. In conclusion, the patient with γY278H did not show dysfibrinogenemia. Moreover, we predicted that γY278H fibrinogen is easier to cleave than γT277R fibrinogen, because γY278H fibrinogen indicated markedly impaired protection against plasmin cleavage compared with γT277R fibrinogen [31].

## 4. Materials and Methods

### 4.1. Patient and Coagulation Tests

PT, APTT, D-dimer, FDPs, functional fibrinogen level, which was assessed using the thrombin time method, and antigenic fibrinogen level, which was assessed using a latex photometric immunoassay with anti-fibrinogen Aα chain monoclonal antibody-coated latex particles (Q-may Laboratory, Oita, Japan), were measured using an automated analyzer, Coapresta2000 (Sekisui Medical, Tokyo, Japan).

### 4.2. DNA Sequence Analysis

Genomic DNA was extracted from white blood cells using a DNA Extraction WB kit (FUJIFILM Wako Pure Chemical, Osaka, Japan) in accordance with the manufacturer’s instructions. To analyze all exons and exon-intron boundaries of the fibrinogen gene, long-range polymerase chain reaction for *FGA*, *FGB*, and *FGG* and direct sequencing were performed as described previously [46].

### 4.3. Immunoblotting Analysis of Plasma Fibrinogen

The patient’s plasma fibrinogen and Aα, Bβ, and γ chains were determined using Western blotting. Purified normal plasma fibrinogen and normal plasma were used as normal controls. The patient’s and normal plasma were diluted 1000 times in *N*-[2-hydroxyethyl] piperazine-*N*’-[2-ethanesulfonic acid] (HEPES), pH 7.4, and 0.12 M NaCl buffer (HBS buffer), separated by SDS-PAGE in non-reducing conditions in a 10% polyacrylamide gel or reducing conditions in a 7% polyacrylamide gel, transferred into a nitrocellulose membrane, and developed with a polyclonal rabbit anti-human fibrinogen antibody (Dako, Carpinteria, CA, USA). The reacting species were visualized using a horseradish peroxidase (HRP) conjugated-goat anti-rabbit IgG antibody (Medical and Biological Laboratories, Nagoya, Japan). Blots were exposed using ECL Western Blotting Detection Reagents (GE Healthcare, Tokyo, Japan) and detected with a ChemiDoc XRS Plus (Bio-Rad, Hercules, CA, USA).

### 4.4. Preparation and Production of Recombinant Fibrinogen Variants

The fibrinogen γ chain expression vector pMLP-γ [51] was altered with six mutagenic primer pairs (Appendix A), and the resultant expression vectors γL276P, γT277R, γT277P, γY278H, γA279D, and γY280C and the wild-type were co-transfected with the histidinol selection plasmid (pMSVhis) into CHO cells that expressed normal human fibrinogen Aα and Bβ chains. Colonies were selected on histidinol (Sigma-Aldrich, St. Louis, MO, USA) and 10–12 cell lines per variant were established as described previously [35]. The highest expressing γT277R or γY278H fibrinogen-producing CHO cell lines were cultured using a roller bottle system. Fibrinogens were purified from the harvested culture medium using ammonium sulfate precipitation followed by immunoaffinity chromatography utilizing a calcium-dependent monoclonal antibody (IF-1, LSI Medience, Tokyo, Japan), as described previously [44]. After elution, fibrinogen was dialyzed, and the purity and characterization of the proteins were determined using SDS-PAGE. Wild-type and γY275C recombinant fibrinogens were used as prepared previously [15].

### 4.5. ELISA

Fibrinogen concentrations in culture media or cell lysates of wild-type and six fibrinogen variants were measured using ELISA as described previously [23]. Briefly, fibrinogen-producing cells were grown to confluence in 60-mm dishes (approximately 3.0 × 10^6^ cells), and the culture media was harvested 1 day after reaching confluence (6–8 days after seeding). Cells were harvested from the same culture dishes, washed 3 times with phosphate-buffered saline (PBS), and finally lysed with 250 µL of 0.1% IGEPAL CA-630 (Sigma-Aldrich) and 10 mM phenylmethylsulfonyl fluoride (Sigma-Aldrich) in 50 mM Tris-HCl buffer pH 8.0. These samples were incubated in 96-well plates, which were pre-coated with a goat anti-human fibrinogen antibody (MP Biomedicals, Irvine, CA, USA), and detected with a HRP conjugated-goat anti-human fibrinogen antibody (MP Biomedicals). The plate was added 3,3′,5,5′-tetramethylbenzidine (SeraCare Life Sciences, Milford, MA, USA) for detection and 1 M phosphoric acid to stop the color development, and absorbance at 450 nm was measured.

### 4.6. Immunoblotting Analysis of Recombinant Fibrinogen

Fibrinogen in the cell lysates of wild-type and six fibrinogen variants were determined as described previously [23,44]. Briefly, fibrinogen-producing cells were grown to confluence in 60-mm dishes, harvested 1 day after reaching confluence (6–8 days after seeding), washed 3 times with PBS, and finally lysed with 75 μL of 5% SDS and 25% glycerol in 250 mM Tris-HCl buffer pH 6.8 (SDS sample buffer). Then, the fibrinogens were determined in the same manner as the immunoblotting analysis of plasma fibrinogen with a polyclonal rabbit anti-human fibrinogen antibody or a monoclonal mouse anti-human fibrinogen γ chain antibody (2G10; Accurate Chemical and Scientific, Carle Place, NY, USA) and a HRP conjugated-goat anti-rabbit IgG antibody or a HRP conjugated-goat anti-mouse IgG antibody (Medical and Biological Laboratories, Nagoya, Japan), respectively.

### 4.7. Fibrinogen Degradation Assays Using Plasmin

Fibrinogen degradation assays of recombinant γT277R and γY278H fibrinogens were performed and compared with those of wild-type and γR275C fibrinogens. Fibrinogen (0.30 mg/mL) in HBS buffer containing 5 mM EDTA were incubated with plasmin (0.18 U/mL; Chromogenix AB, Molngal, Sweden) for 0, 0.5, 1, 2, or 4 h at 37 °C. Reactions were stopped by the addition of SDS sample buffer followed by boiling for 5 min. The plasmin digestion products were separated by SDS-PAGE on 10% polyacrylamide gel and stained with CBB R-250.

### 4.8. Thrombin-Catalyzed Fibrin Polymerization

Fibrin polymerization of recombinant γT277R and γY278H fibrinogens catalyzed by human α-thrombin (Enzyme Research Laboratories, South Bend, MA, USA) was performed as described previously [46]. Briefly, fibrinogen (90 µL at 0.20 mg/mL) in 20 mM HBS buffer was mixed with thrombin (10 µL at 0.5 U/mL) at room temperature. The turbidity change at 350 nm was followed using a UVmini-1280 spectrophotometer (Shimadzu, Tokyo, Japan), and reactions were performed in triplicate for each sample. For reaction parameters, the lag time and maximum slope were obtained from turbidity curves. Wild-type and γR275C fibrinogens were used to compare these functions.

### 4.9. Protection Assay for the Plasmin Digestion of Fibrinogen

Protection assays for the plasmin digestion of recombinant γT277R and γY278H fibrinogens were performed as described previously [46]. Briefly, 0.30 mg/mL fibrinogen in HBS buffer containing 1 or 5 mM CaCl_2_ or 5 mM EDTA was incubated with plasmin (0.18 U/mL) for 2 h at 37 °C. The reactions were stopped by adding SDS sample buffer followed by boiling for 5 min. The plasmin digests were then separated by SDS-PAGE on 10% polyacrylamide gel and stained with CBB R-250. Wild-type and γR275C fibrinogens were used to compare these functions.

### 4.10. FXIIIa-Catalyzed Cross-Linking of Fibrinogen

FXIIIa-catalyzed cross-linking (γ-γ dimer formation) of γT277R and γY278H purified fibrinogen was performed as described previously [46]. Briefly, fibrinogen (0.47 mg/mL) was incubated at 37 °C with FXIII (66.5 U/mL; Enzyme Research Laboratories), which was activated with thrombin at 37 °C for 1 h in HBS buffer with CaCl_2_, in the presence of hirudin (3.3 U/mL; Sigma-Aldrich) as a thrombin inhibitor. The reactions were stopped by adding SDS sample buffer followed by boiling for 5 min, and the products were separated by SDS-PAGE in reducing conditions in an 8% polyacrylamide gel and stained with CBB R-250. Wild-type and γR275C fibrinogens were used to compare these functions.

### 4.11. Statistical Analysis

Statistical analysis was performed with EZR software (Saitama Medical Center, Jichi Medical University, Saitama, Japan), which is a graphical user interface for R software (The R Foundation for Statistical Computing, Vienna, Australia). The differences in fibrinogen production measured by ELISA were assessed using the Kruskal–Wallis test and Steel–Dwass test, and the parameters of thrombin-catalyzed fibrin polymerization were assessed using one-way analysis of variance (ANOVA) and the Dunnett’s test. A difference was considered to be significant when *p* < 0.05.

## 5. Conclusions

We identified a novel heterozygous variant, γY278H, designated as Hiroshima. There was inconsistency between the plasma fibrinogen level and the production in fibrinogen-producing CHO cells. The patient showed hypofibrinogenemia, but the secretion by fibrinogen-producing CHO cells was not reduced. We predicted that the fibrinogen production ability of fibrinogen-producing cell lines reflected patients’ plasma fibrinogen levels from the results of our established 33γ-module variant fibrinogen-producing CHO cell lines. Our investigation demonstrated that γY278H fibrinogens were easily cleaved by plasmin. We concluded that this patient with γY278H showed hypofibrinogenemia because γY278H fibrinogen was secreted normally from the patient’s hepatocytes but then underwent accelerated degradation by plasmin in the circulation. Moreover, γT277R also showed that the patients’ plasma fibrinogen levels were reduced but secretion by fibrinogen-producing CHO cells were not reduced for the same reason as γY278H.

## Figures and Tables

**Figure 1 ijms-22-05218-f001:**
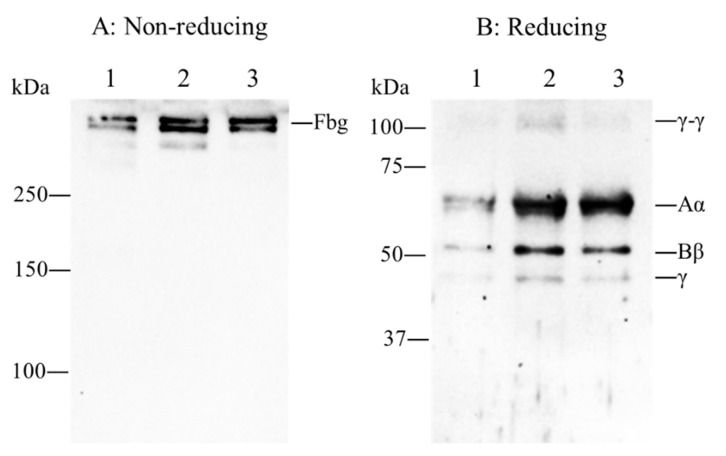
Immunoblotting analysis of plasma fibrinogen. One thousand-fold diluted plasma was separated by sodium dodecyl sulfate (SDS)-polyacrylamide gel electrophoresis (PAGE) in non-reducing conditions in a 7% polyacrylamide gel (**A**) or reducing conditions in a 10% polyacrylamide gel (**B**) and plasma fibrinogen was detected by Western blotting using an anti-human fibrinogen antibody. Fibrinogen (Fbg), γ-γ dimer and Aα, Bβ, and γ chains are indicated on the right side of each panels. Lane 1: purified normal plasma fibrinogen, 2: normal plasma, 3: patient’s plasma.

**Figure 2 ijms-22-05218-f002:**
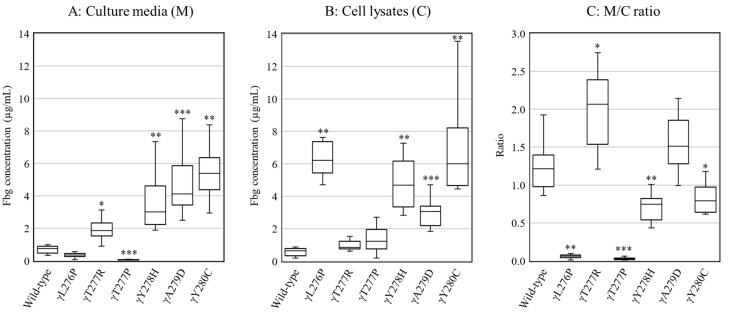
Secretion and synthesis of fibrinogen in Chinese hamster ovary cells. Fibrinogen concentrations in the culture media (**A**) and cell lysates (**B**) were measured using an enzyme-linked immunosorbent assay. Panel (**C**) shows the M/C ratio of the culture media to the cell lysate. Box plots and central bars show the interquartile range and median, respectively. The significance of differences between wild-type and variant fibrinogen-producing cells is shown (* *p* < 0.05, ** *p* < 0.01, *** *p* < 0.001).

**Figure 3 ijms-22-05218-f003:**
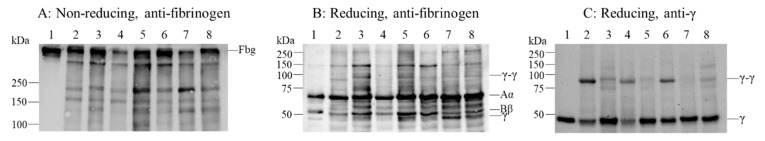
Immunoblotting analysis of recombinant fibrinogen. Fibrinogen in Chinese hamster ovary cell lysates were determined by Western blotting using anti-human fibrinogen antibodies in non-reducing conditions in a 8% polyacrylamide gel (**A**) or reducing conditions in an 10% polyacrylamide gel (**B**) and anti-human fibrinogen γ-chain antibodies in reducing conditions in an 10% polyacrylamide gel (**C**). Fibrinogen (Fbg), γ-γ dimer and Aα, Bβ, and γ chain are indicated on the right side of each panel. Lane 1: purified fibrinogen, 2: wild-type, 3: γL276P, 4: γT277R, 5: γT277P, 6: γY278H, 7: γA279D, and 8: γY280C cell lines.

**Figure 4 ijms-22-05218-f004:**
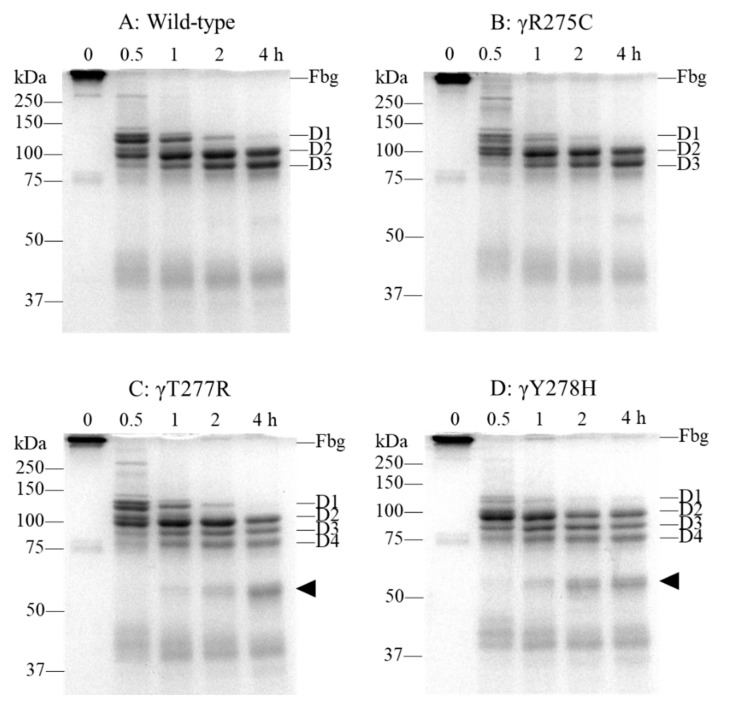
Fibrinogen degradation assay with plasmin. Recombinant fibrinogens (0.30 mg/mL) in HBS buffer containing 5 mM EDTA were incubated with 0.18 U/mL plasmin for 0–4 h at 37 °C. The fragments were analyzed by 10% SDS-PAGE and with Coomassie brilliant blue (CBB) staining. Fibrinogen (Fbg), fragments D1, D2, D3, and D4, and lower molecular weight fragment (◀) are indicated on the right side of each panel. (**A**) wild type, (**B**) γR275C, (**C**) γT277R, (**D**) γY278H.

**Figure 5 ijms-22-05218-f005:**
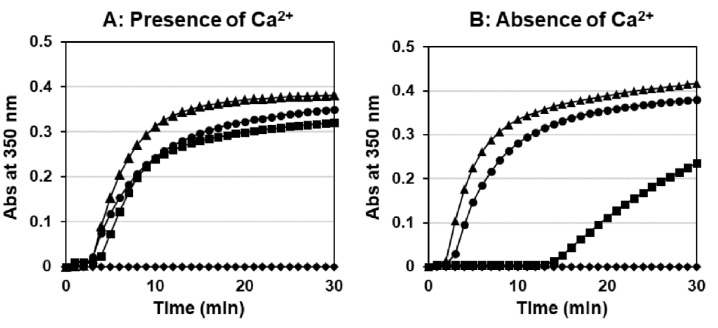
Thrombin catalyzed fibrin polymerization. The polymerization of 0.18 mg/mL recombinant fibrinogens was initiated with 0.05 U/mL thrombin in the presence of 1 mM Ca^2+^ (**A**) or absence of Ca^2+^ (**B**). ●: Wild-type, ♦: γR275C, ▲: γT277R, ■: γY278H.

**Figure 6 ijms-22-05218-f006:**
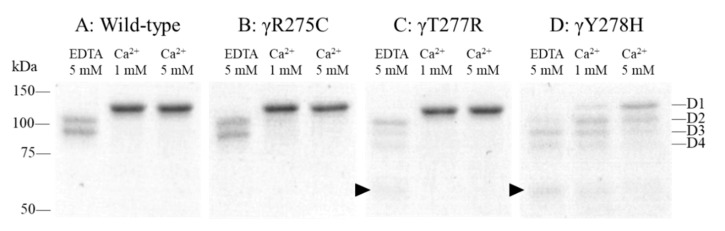
Protection assay for plasmin degradation of fibrinogen. Recombinant fibrinogen (0.30 mg/mL) in HBS buffer containing 5 mM EDTA or 1 or 5 mM Ca^2+^ was incubated with 0.18 U/mL plasmin at 37 °C for 2 h. The fragments were analyzed using 10% SDS-PAGE with CBB staining. Fragments D1, D2, D3, and D4 are indicated on the right side of panel D and lower molecular weight fragments are indicated (►). (**A**) wild type, (**B**) γR275C, (**C**) γT277R, (**D**) γY278H.

**Figure 7 ijms-22-05218-f007:**
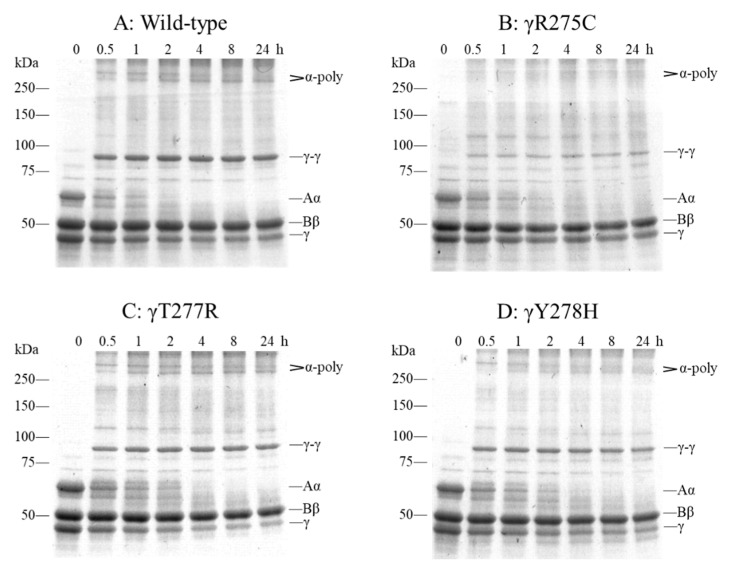
FXIIIa-catalyzed cross-linking of fibrinogen. Recombinant fibrinogens were cross-linked using FXIIIa for 0.5–24 h and examined in 8% SDS-PAGE gels in reducing conditions with CBB staining. Reduced fibrinogen chains (Aα, Bβ, γ chains, α-polymer, γ-γ dimer) are indicated on the right side of each panel. (**A**) wild type, (**B**) γR275C, (**C**) γT277R, (**D**) γY278H.

**Table 1 ijms-22-05218-t001:** Recombinant fibrinogen production and patients’ plasma fibrinogen level. We found the production of recombinant fibrinogens from Chinese hamster ovary cells was reduced when the ratio in culture media to cell lysate (M/C ratio) was <0.5 (grey), and the patients’ plasma fibrinogen levels were reduced when their antigenic fibrinogen level was <1.5 g/L (grey). The agreement or disagreement between recombinant fibrinogen production and plasma fibrinogen level is indicated as ◯ or ✕, respectively. NA; not applicable. *: categorized as authors’ description.

Variant	Recombinant Fibrinogen Production		Patients’ Plasma Fibrinogen Level		Disagreement
	M/C (M/C of Wild-Type)	Reference	Antigenic (g/L)	Reference	
γC153R	NA (2.09 ± 1.38)	[23]	0.87	[23]	◯
γR275C	Not reduced	[15]	2.91	[17] and 26 cases	◯
γR275H	Not reduced	[24]	2.02	[17] and 31 cases	◯
γT277P	0.03 ± 0.02 (1.24 ± 0.31)	This manuscript	1.18	[21]	◯
γT277R	2.03 ± 0.50 (1.24 ± 0.31)		0.79	[22]	✕
γY278H	0.72 ± 0.17 (1.24 ± 0.31)		1.16	This manuscript	✕
γA279D	1.62 ± 0.48 (1.24 ± 0.31)		2.7	[25]	◯
γY280C	0.82 ± 0.20 (1.24 ± 0.31)		NA *	[20] and 1 case	◯
γG284R	0.05 ± 0.01 (0.77 ± 0.22)	[24]	1.00	[19]	◯
γA289V	0.07 ± 0.02 (0.88 ± 0.17)	[26]	0.50	[26] and 1 case	◯
γG292V	0.35 ± 0.12 (0.88 ± 0.17)	[26]	1.20	[27]	◯
γT305A	1.12 ± 0.14 (1.66 ± 0.40)	[28]	1.12	[28]	✕
γH307Y	0.31 ± 0.09 (1.66 ± 0.40)	[28]	0.86	[29]	◯
γN308K	1.74 ± 0.62 (1.66 ± 0.40)	[28]	2.46	[30] and 4 cases	◯
γN308I	Not reduced	[31]	NA *	[32]	◯
γS313N	0.06 ± 0.01 (1.05 ± 0.17)	[18]	0.58	[33]	◯
γT314P	0.08 ± 0.04 (0.77 ± 0.22)	[24]	0.47	[34]	◯
γD316N	0.26 ± 0.05 (0.77 ± 0.22)	[24]	1.36	[21] and 1 case	◯
γD318Y	0.70 ± 0.17 (1.30 ± 0.19)	[35]	2.9	[36]	◯
γΔ319	0.03 ± 0.01 (1.30 ± 0.19)	[35]	NA *	Czwalinna A, 2004 [12] (on-line submission)	◯
γN319K	0.05 ± 0.01 (1.30 ± 0.19)	[35]	NA	Meyer M, 2000 [12] (congress abstract)	NA
γΔ320	0.09 ± 0.03 (1.30 ± 0.19)	[35]	0.80	[37] and 1 case	◯
γD320E	0.06 ± 0.03 (1.30 ± 0.19)	[35]	1.30	[38]	◯
γD320G	0.11 ± 0.02 (1.30 ± 0.19)	[35]	0.85	[35] and 1 case	◯
γC326S	0.03 ± 0.02 (2.17 ± 0.57)	[39]	0.81	[40] and 3 cases	◯
γC326Y	0.02 ± 0.03 (2.17 ± 0.57)	[39]	1.50	[41] and 1 case	◯
γM336I	0.05 ± 0.01 (1.05 ± 0.17)	[18]	NA *	[41] and 1 case	◯
γA341D	0.15 ± 0.06 (1.05 ± 0.17)	[18]	1.37	[42]	◯
γN345D	0.09 ± 0.02 (1.05 ± 0.17)	[18]	<0.5	[41]	◯
γD364H	Not reduced	[43]	3.40	[22]	◯
γG366S	0.03 ± 0.02 (0.77 ± 0.22)	[24]	0.95	[21]	◯
γR375G	1.10 ± 0.25 (0.94 ± 0.10)	[44]	2.50	[45]	◯
γR375W	0.09 ± 0.10 (0.94 ± 0.10)	[44]	1.10	13 cases	◯

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
