# Peer review of "Recombinant γY278H Fibrinogen Showed Normal Secretion from CHO Cells, but a Corresponding Heterozygous Patient Showed Hypofibrinogenemia"

_ijms, 2021, doi:10.3390/ijms22105218_

Round 1

Reviewer 1 Report

In this paper the Authors report a novel gamma-chain variant of fibrinogen in a patient heterozygous for the mutation.. They expressed the mutation and found a normal synthesis and release. However, the mutation seemed to induce and increased susceptibility to the action of plasmin.  

Comments

  • The title is not informative and the increased susceptibility to plasmin degradation should be included
  • Because of the supported view of an accelerated degradation by plasmin of the abnormal fibrinogen molecule, the Authors should report data on D-dimer and FDPs measurements in patient’s plasma. Furthermore, to complete the laboratory phenotype, data on thrombin time and reptilase time would be useful
  • All the expression experiments have been performed by using “homozygous” expression, no data on co-expression wild-type and gammaY287H mutation has been provided. This is important because the patient is heterozygous and one would know whether co-expression induces the secretion of only the normal molecule or only the mutated molecule. According to the Authors, hypofibrinogenemia is caused by accelerated plasma degradation by plasmin, but the patient is heterozygous and thus probably the in vitro model is not completely matching the phenotype. On would know whether both normal and abnormal molecules are secreted when co-expressed
      •  

Author Response

Reviewer 1

Thank you so much for reviewing and some useful comments. We revise our manuscript as follows.

The title is not informative and the increased susceptibility to plasmin degradation should be included

          We do not describe about plasmin degradation in the title, because our results suggest that γY278H fibrinogen is accelerated degradation by plasmin but did not demonstrate FDPs is presence in plasma of the patient.

Because of the supported view of an accelerated degradation by plasmin of the abnormal fibrinogen molecule, the Authors should report data on D-dimer and FDPs measurements in patient’s plasma. Furthermore, to complete the laboratory phenotype, data on thrombin time and reptilase time would be useful

       We assessed the patient’s D-dimer (<0.1 mg/L) and FDPs (<2.5 mg/L), and described in page 2, line 81. Contrary to our expectations, not only D-dimer but also FDPs were not detected by coagulation tests and immunoblotting analysis (Figure 1). We suppose that the reason is plasmin degradation products were markedly smaller and these were easier to cleaved than fibrinogen.

       Because we have small amount of patient’s plasma to perform the thrombin- or reptilase clotting time, we prepared recombinant fibrinogens and performed some functional tests. Although we could not perform the functional test of patient’s fibrinogen, we suppose that fibrin polymerization of patient’s fibrinogen was almost normal because functional fibrinogen level was almost same as antigenic fibrinogen level.

All the expression experiments have been performed by using “homozygous” expression, no data on co-expression wild-type and gammaY287H mutation has been provided. This is important because the patient is heterozygous and one would know whether co-expression induces the secretion of only the normal molecule or only the mutated molecule. According to the Authors, hypofibrinogenemia is caused by accelerated plasma degradation by plasmin, but the patient is heterozygous and thus probably the in vitro model is not completely matching the phenotype. On would know whether both normal and abnormal molecules are secreted when co-expressed

       If we establish co-expressed fibrinogen-producing CHO cells, we cannot distinguish actual amount of produced wild-type and mutated fibrinogen. Namely, it is difficult to establish CHO cell lines which produce similar amount of wild-type and mutated fibrinogen.

We have already demonstrated that both normal and mutated fibrinogens were present in patients’ plasma in many heterozygous CFD cases. In general, three types of dimers (normal-normal, normal-mutated, mutated-mutated) are present in a 1:2:1 ratio in the plasma of heterozygous patient. In the case of heterozygous hypofibrinogenemic variant, we predict one-fourth of fibrinogen, namely, only normal-normal dimers are present in blood circulation. Finally, our results suggest that in heterozygous γY287H fibrinogen patient normal-mutated and mutated-mutated dimers are accelerated in degradation, and resulting in hypofibrinogenemia.

Reviewer 2 Report

The authors identified and characterized a new genetic mutation in a patient with mild hypofibrinogenemia in the D domain of the γ chains. It is very important to identify new genetic variants because it gives the possibility to confirm the diagnosis, to elaborate a diagnostic strategy, to distinguish in some cases that the patient is at risk of thrombosis rather than bleeding,

The methodological part is excellently processed using the best and latest methods, which are key to the characterization of new genetic variants in patients with congenital hypofibrinogenemia.

Figures and tables in the text are very clearly written.

I have to say that with these 48 references there are only very few references newer than 5 years old. Therefore, it is necessary to add newer references.

Major comments

-Introduction, page 1, lines 36-44 - The characteristics of fibrinogen are also described in the manuscript, which should be cited. At the same time, it would be a newer citation than stated by the authors: ,, Simurda T. et al. Fibrinogen Martin: A Novel Mutation in FGB (Gln180Stop) Causing Congenital Afibrinogenemia. Seminars in Thrombosis and Hemostasis. 2016 Jun;42(4):455-458. DOI: 10.1055/s-0036-1581104.“

-Introduction, page 2, lines 61-68 Authors should more accurately determine the total number of fibrinogen variants. The review, which focused on genetic variants in the FGB and FGG genes coding for the fibrinogen D-domain in congenital quantitative fibrinogen disorders, stated that the GEHT database reports 1,215 molecular abnormalities of fibrinogen (626 in the FGA gene, 154 in the FGB gene, and 435 in the FGG gene). This was published in a manuscript which should be cited:. ,, Simurda, T et al. Genetic Variants in the FGB and FGG Genes Mapping in the Beta and Gamma Nodules of the Fibrinogen Molecule in Congenital Quantitative Fibrinogen Disorders Associated with a Thrombotic Phenotype. Int. J. Mol. Sci. 2020, 21, 4616. https://doi.org/10.3390/ijms21134616“

- page 2, line 68: The authors forgot to add a citation, it is appropriate to cite a review of congenital fibrinogen disorders, which describes the characteristics of congenital fibrinogen disorders.,, de Moerloose P et al. Congenital fibrinogen disorders: an update. Semin Thromb Hemost. 2013 Sep;39(6):585-95. doi: 10.1055/s-0033-1349222.“

-Results, page 2, lines 79-83 The authors should describe the clinical phenotype of the patient in more detail, any other bleeding or thrombotic complications were observed. It is very important in fibrinogen disorders research to detect genotype and phenotype correlations. This localization is often associated with a thrombotic phenotype, so the clinical description is important.

-Discussion, page 7, lines 215-219. In addition to the mentioned genetic mutations, a novel nonsense mutation γ-p.Glu275Stop was identified in this localization, which was named Fibrinogen Martin III, in which protein modeling was performed. The manuscript clearly states that γ chain and the C-terminal domain are critical for fibrinogen secretion from hepatocytes. With these premises, it is highly probable that in fibrinogen molecule bearing the γ-p.Glu275Stop variant could not be competent for secretion. Authors should quote this: Simurda, T. et al. Congenital hypofibrinogenemia associated with a novel heterozygous nonsense mutation in the globular C-terminal domain of the γ-chain (p.Glu275Stop). J Thromb Thrombolysis 50, 233–236 (2020). https://doi.org/10.1007/s11239-019-01991-x

Author Response

Reviewer 2

Thank you so much for reviewing and some useful comments. We revise our manuscript as follows.

Introduction, page 1, lines 36-44

Introduction, page 2, lines 61-68 and line 68

          Thank you for your advice and we add introduced references (Reference 4, 13, 14).

Results, page 2, lines 79-83 The authors should describe the clinical phenotype of the patient in more detail, any other bleeding or thrombotic complications were observed. It is very important in fibrinogen disorders research to detect genotype and phenotype correlations. This localization is often associated with a thrombotic phenotype, so the clinical description is important.

          We describe additional information of the patient in page 2, lines 80-81.

Discussion, page 7, lines 215-219. In addition to the mentioned genetic mutations, a novel nonsense mutation γ-p.Glu275Stop was identified in this localization, which was named Fibrinogen Martin III, in which protein modeling was performed. The manuscript clearly states that γ chain and the C-terminal domain are critical for fibrinogen secretion from hepatocytes. With these premises, it is highly probable that in fibrinogen molecule bearing the γ-p.Glu275Stop variant could not be competent for secretion.

       In our manuscript, we discuss about only missense variants nearby γY278 (p.γY304), so we only quote references [17]-[22].

Round 2

Reviewer 2 Report

The presented manuscript has been corrected in response to the suggestions. The authors have followed the recommendations of the reviewer. After the revision, the provided data and addition of the results became more clear.  I would like to thank the authors for resubmitting the manuscript and explaining the obscure points from the previous version.